# Chondrosarcoma Resistance to Radiation Therapy: Origins and Potential Therapeutic Solutions

**DOI:** 10.3390/cancers15071962

**Published:** 2023-03-24

**Authors:** Antoine Gilbert, Mihaela Tudor, Juliette Montanari, Kevin Commenchail, Diana Iulia Savu, Paul Lesueur, François Chevalier

**Affiliations:** 1UMR6252 CIMAP, Team Applications in Radiobiology with Accelerated Ions, CEA—CNRS—ENSICAEN—Université de Caen Normandie, 14000 Caen, France; 2Department of Life and Environmental Physics, Horia Hulubei National Institute of Physics and Nuclear Engineering, 077125 Magurele, Romania; 3Faculty of Biology, University of Bucharest, 050095 Bucharest, Romania; 4Radiation Oncology Department, Centre François Baclesse, 14000 Caen, France; 5Radiation Oncology Department, Centre Guillaume Le Conquérant, 76600 Le Havre, France; 6ISTCT UMR6030-CNRS, CEA, Université de Caen-Normandie, Equipe CERVOxy, 14000 Caen, France

**Keywords:** chondrosarcoma, radiation resistance, hadrontherapy, carbon ions therapy, hypoxia, cancer stem cells

## Abstract

**Simple Summary:**

This review article aims in describing the origins of chondrosarcoma radiation-resistance and proposes several potential solutions in order to improve the treatment, in regard with tumor grade and characteristics.

**Abstract:**

Chondrosarcoma is a malignant cartilaginous tumor that is particularly chemoresistant and radioresistant to X-rays. The first line of treatment is surgery, though this is almost impossible in some specific locations. Such resistances can be explained by the particular composition of the tumor, which develops within a dense cartilaginous matrix, producing a resistant area where the oxygen tension is very low. This microenvironment forces the cells to adapt and dedifferentiate into cancer stem cells, which are described to be more resistant to conventional treatments. One of the main avenues considered to treat this type of tumor is hadrontherapy, in particular for its ballistic properties but also its greater biological effectiveness against tumor cells. In this review, we describe the different forms of chondrosarcoma resistance and how hadrontherapy, combined with other treatments involving targeted inhibitors, could help to better treat high-grade chondrosarcoma.

## 1. Introduction

### 1.1. Overview

Chondrosarcoma is a malignant cartilage tumor of the bone that accounts for about 20–30% of all primary bone sarcomas. This tumor generally affects adults between the ages of 30 and 60 and develops in the limb cavity or on the surface of the bone [1,2,3]. Chondrosarcomas are uncommon in children and adolescents and represent less than 5% of total chondrosarcomas [4]. Chondrosarcoma can develop de novo or from benign cartilage tumors of the bone such as osteochondromas and enchondromas [1]. Anatomically, it is located in the pelvic area, most frequently the ilium, followed by the proximal femur, proximal humerus, distal femur, and ribs. The symptoms can last a long period of time, from months to years, and include pain, pathologic fracture, and common lung metastasis [5]. The number of reported cases in the skull is low; tumors in the nasal cavity are especially rare. This type of chondrosarcoma is more common in children than in adults, with the youngest reported patient aged two years [6]. Due to the location, complete resection is difficult to obtain, and a recurrence of the tumor is quite common.

According to the 2020 World Health Organization, chondrosarcomas are grouped in the malignant category and are classified into three main grades I–III, based on cellularity, tumor matrix characteristics, nuclear features, and mitotic rate [7,8]. Grade I chondrosarcomas are comprised of few cells with no variation in shape and size, and a low metastatic potential associated with flat bone. Both grade II and III chondrosarcomas present hyper-cellularity, variation in cell morphology, and invasion of surrounding tissue. Grade II chondrosarcoma presents a myxoid component while grade III chondrosarcoma is characterized by intense mitotic activity [7,8,9,10].

Chondrosarcomas can also be divided into three subtypes: central, peripheral, and periosteum. The most common type, which happens in about 70% of cases in the proximal femur or proximal humerus, is central chondrosarcoma. The long bones, pelvis, and scapular belt are usually affected by peripheral chondrosarcoma and can develop from pre-existing osteochondromas while periosteal chondrosarcoma can develop on the surface of the bone [1,11,12]. Chondrosarcoma can be classified into lesser-known subtypes such as dedifferentiated, mesenchymal, clear cell, and extra-skeletal myxoid chondrosarcoma [13]. Dedifferentiated chondrosarcomas develop from lower-grade common chondrosarcoma, while mesenchymal chondrosarcoma is very malignant and has a biomorphic histological model with small, round cell islands that interpenetrate with cartilage and dedifferentiate spindle cells [13,14]. Clear-cell chondrosarcoma is less aggressive and is composed of cells containing a large amount of glycogen in the cytoplasm [13,15]. Extra-skeletal myxoid chondrosarcoma is a soft extraskeletal tissue with uncertain differentiation, a predilection for extremities, and a low growth rate [13,16].

### 1.2. Genetic Characteristics of Chondrosarcoma

Chondrosarcoma relapses and metastases often, thus it is important to identify biomarkers that determine the best clinical approach. Diagnosing chondrosarcomas can be challenging due to its rather common symptoms, low incidence, and grading system [17]. As such, making a correct diagnosis is a key factor for the overall treatment of the tumor. Recent development in imaging methods, endoscopic techniques, gene analysis, biomarker detection, and immunological, and surgical approaches have reduced diagnostic delays [18]. The most commonly used method of diagnostic is radiography, though other imaging methods such as tomography scan, magnetic resonance imaging, bone scintigraphy, and positron emission tomography have been used as adjuvants for patient evaluation [19].

The most frequent mutations found in chondrosarcomas are on isocitrate dehydrogenase (IDH) genes, on arginines R132 for IDH1 and R140/R172 for IDH2 [20]. These genes codes for Krebs cycle enzymes are responsible for the conversion of isocitrate into α-ketoglutarate (α-KG). These mutations induce a gain in function of these enzymes, which can then transform α-KG into an oncometabolite, D-2-hydroxyglutarate (D-2-HG) [21]. Decreased α-KG and increased D-2-HG are associated with epigenetic modifications, such as decreased DNA methylation and hypermethylation of histones associated with differentiation [22]. These changes can also impact the microenvironment since they can impact collagen maturation [23] and block the activity of prolyl hydroxylases [24] thus inducing the stabilization of HIF-1α [25] and HIF-2α [26], responsible for adaptation to hypoxia. The expression of these factors, independently of O2 concentrations, will be responsible for tumor progression and radioresistance. In addition, chondrosarcoma cells are often EXT1/2 mutated and CDKN2A/B deletions are also frequent, as well as COL2A1 mutations [27]. In osteosarcoma, another rare malignant bone tumor, the expression of several repetitive elements was observed differentially expressed with normal bone. HERVs’ (human endogenous retroviruses) integrated sequences and satellite elements were the most significantly differential expressed elements between osteosarcoma and normal tissues and could help to understand the genesis mechanism of such sarcoma [28]. In addition, a transcriptomic analysis of osteosarcoma bone samples revealed that BTNL9, MMP14, ABCA10, ACACB, COL11A1, and PKM2 were expressed differentially with the highest significance between tumor and normal bone, and reflected the changes in the regulation of the degradation of collagen and extracellular matrix [29].

### 1.3. Chondrosarcoma Standard Treatment

According to ESMO guidelines [30], standard treatment depends on the localization, the histological subtype, and the level of differentiation. Grade I low-grade peripheral chondrosarcoma (arising from osteochondroma) or grade I central chondrosarcoma of the long bones, can be managed with minimally-invasive surgery (curetage for example) without adjuvant treatment. On the opposite higher-grade chondrosarcoma, such as clear cell chondrosarcoma, mesenchymal chondrosarcoma, and dedifferentiated or axial chondrosarcoma (pelvic, spine, or skull base) should be extensively resected with wide margins and postoperative irradiation is often proposed in case R1 or R2 resections, or for the management of local recurrences. For R1 and R2 low-grade skull-base chondrosarcomas, the best timing for irradiation remains to be discussed. Immediate or delayed irradiation could be proposed according to each center’s policies. Recent evidence suggests that mesenchymal chondrosarcoma may be chemotherapy sensitive, and may be considered for adjuvant or neoadjuvant therapy [30]. The next parts of this review article will be specifically focused on high-grade chondrosarcoma.

It is known that chondrosarcoma is a chemo- and radioresistant tumor [31]. Slow proliferation, overexpression of the protein involved in drug resistance MDR1, poor vascularization, and dense extracellular matrix may be responsible for chemo- and radioresistance of chondrosarcoma [32]

Although many studies have obtained good results, monotherapy of heterogeneous chondrosarcoma is a concern due to the tumor’s ability to adapt. Moreover, the failure of a monotherapy should not exclude its potential for being an adjuvant therapy in combined treatment. The use of novel nonconventional therapies can improve the outcome, though the adverse effects should be taken into consideration and further clinical investigations are required to assess the safety and efficacy in a large group of patients.

#### 1.3.1. Chondrosarcoma Chemotherapy

Chemotherapy is rarely effective and the studies on patients are limited due to the rarity of these diseases [32]. As such, current treatments have a base in ostosarcoma treatment [33]. As such, palliative treatments with cisplatin, doxorubicin, or ifosfamide are also used in clinical treatments [34] even though chondrosarcoma has presented resistance to doxorubicin in vitro [35].

Nonconventional treatments for chondrosarcoma include molecularly-targeted therapies, epigenetic approaches, immunotherapy, and herbal therapies [33,36]. Some of the targets for chondrosarcoma therapies involve mutations of isocitrate dehydrogenases IDH1 and IDH2 [37,38,39,40], angiogenesis [41,42], cyclin-dependent kinases (CDK) [43], tyrosine kinase inhibitor [44], mechanisms involved in the signaling pathway of Rapamycin (mTOR) [45,46], agents of hypomethylation, and histone deacetylase (HDAC) [47,48,49]. Clinical studies involving immune checkpoint inhibition are in the early stages and show promising results despite patients’ mixed responses to some inhibitors [50,51,52]. A large and complete overview of new targeted therapies in chondrosarcoma can be found in two recent review studies by Boehme et al. [32] and Tlemsani et al. [27].

#### 1.3.2. Chondrosarcoma Radiotherapy

Conventional radiation therapy is used for patients with incomplete resections, inoperable tumors, or metastases [53,54,55].

Photon radiotherapy has low accuracy, with undesirable toxicity in the surrounding normal tissue, and is not suitable for targeting tumors larger than 2.5 cm [56]. Over the past decade, radiation therapy has developed better control over localization and dosage with a limited effect on surrounding healthy tissue [57]. There are several types of radiological techniques used to treat inoperable tumors such as external radiation/source therapy, modified fraction radiotherapy, internal radiation therapy, and particle therapy. Therefore, each type of radiation therapy machine has distinct physical characteristics, which may influence clinical outcomes [58].

Clinical studies have shown poor results from low-dose radiation therapy. Indeed, delivering a dose higher than 70 Gy is mandatory to obtain local control. This dose is difficult to reach with conventional radiotherapy due to surrounding neural structures for which the tolerance dose is well below 70 Gy (from 54 to 60 Gy for the optic tract or brainstem, and 50 Gy for the spinal cord, for example). Modern techniques such as intensive modulated radiotherapy (IMRT), stereotactic radiosurgery, and hadron therapy can overcome these limits [59]. Reports show long-term promising results in patients with spine tumors when using a combination of IMPT and IMRT [60] or for patients with intracranial chondrosarcoma undergoing stereotactic radiosurgery [61]. Survival rates depend on the dose, age, tumor size, and quality of the surgical treatment. An improvement in radiation delivery remains necessary to increase the therapeutic ratio [55].

## 2. Radiation Resistance of Chondrosarcoma: Microenvironment, Molecular and Cellular Consequences

As previously mentioned, chondrosarcoma is particularly resistant to conventional radiotherapy, especially due to its very dense cartilaginous extracellular matrix and the presence of some cells in the tumor tissue that may proliferate slowly, whereas radiotherapy (RT) is more effective on rapidly dividing cells [62].

### 2.1. Hypoxia-Related Radiation Resistance

The characteristics mentioned above, and the low proportion of blood vessels in these tumors, also lead to a hypoxic microenvironment in the tumor. It is widely described that the decrease in oxygen content causes a reduction in the effectiveness of X-ray RT. Indeed, the lethal effects of X-rays are triggered by indirect DNA damage, which is mainly caused by the formation of reactive oxygen species (ROS) due to the radiolysis of water and dissolved oxygen [63]. Thus, hypoxia leads directly to a decrease in the efficiency of RT via an absence of O_2_ concentrations (Figure 1).

Hypoxia allows an adaptation of the cancer cells which can give rise to radioresistance. Different signaling pathways have been described, such as autophagy increase [64], stabilization and signaling of the HIF-1 factor, increased secretion of exosomes, or reprogramming of energy metabolism [65].

Concerning chondrosarcoma, HIF-1α stabilization is observed when cells are cultured in hypoxia [66]. The expression of HIF-1α notably allows the activation of angiogenesis pathways, via the expression of vascular endothelial growth factor (VEGF) [66,67]. In the clinic, the high expression of the hypoxia factor HIF1-α is notably associated with poor prognosis and metastatic tumors with poor patient survival. This suggests that activation of the transcription factor HIF-1α may play a role in tumor progression [68,69]. More recent studies have shown that HIF-2α also plays an important role in the progression of chondrosarcoma cells by promoting tumor-initiating and invasive properties [26]. In addition, mutations found in chondrosarcoma such as mutations located in IDH1/2 (>50% of chondrosarcomas) have been described to induce constitutive activation of HIF-1α and Hif-2α factors [25,26]. Although hypoxia is known to induce radioresistance in other cancer cell models, very few publications have been able to make this link directly in chondrosarcoma. For example, one study showed that overexpression of HIF-1α was associated with an overexpression of BCL-xl (a bcl2 family factor), notably conducting to antiapoptotic properties that can generate chemo- and radioresistance [70]. In closely related models, such as osteosarcoma, it has been shown that hypoxia can induce increased autophagy in connection with radioresistance [64].

The hypoxic microenvironment can also cause a form of resistance, by favoring the presence of more resistant cellular subtypes, the cancer stem cells (CSCs) [71]. Indeed, it has been described that the factors HIF-1α and HIF-2α allow the proliferation of CSCs via the activation of the PI3K/AKT [72] and NF-κB [73] pathways. These hypoxia factors enable the activation of the notch [74,75], and hedgehog [76] pathways, responsible for maintaining the stem potential of this cell subtype [77].

### 2.2. Radioresistance Links with CSCs

Cancer stem cells (CSCs) are a subpopulation of cancer cells within the tumor that have been associated with treatment resistance, tumor relapse, and metastasis in several cancers including chondrosarcoma (CS). CSC (or tumor-initiating cells) are seen as drivers of tumor establishment and growth, often correlated to aggressive, heterogeneous, and therapy-resistant tumors [78]. The concept of CSC is also related to specific cellular biomarkers. Indeed, several markers expressed in CSCs can also be found in adult tissue-resident stem cell populations [79]. In order to describe this particular type of cell, we kept the term of CSCs, knowing that it could be the subject of controversial theories [80,81]. CSCs are defined as dedifferentiated cells that have unlimited proliferation and self-renewal abilities and can reinitiate and reconstitute tumor heterogeneity [82]. CSCs have inherited normal stem cells properties, including a hypoxic niche that protects them from treatments and promotes the quiescence state, telomerase activity, an increase in the activity of membrane transporters and detoxification enzyme, activation of antiapoptotic pathways, an enhanced DNA repair capacity, and catabolism of ROS (Figure 1). These properties will block or reduce the cellular effects induced by actual antitumor treatments [2,3]. These persisting cells, even few in number, can proliferate and reconstitute the tumor with all its phenotypic diversity. Resistance to therapeutic treatment, such as chemotherapy and radiotherapy, could be associated with the fact that current therapies do not target CSCs [83].

Several CSC identification and isolation methods have been described [84]. In vitro assays are commonly used to isolate CSCs such as the sphere formation assay called tumorspheres, under nonadherent, serum-free conditions and enriched with growth factors. In vivo, the isolated CSCs are transplanted in immunocompromised mice to assess the tumorigenic capacity of the CSCs. As CSCs have the same properties as normal stem cells, they can be identified within the tumors with normal stem cell markers such as SOX2, OCT3/4, Nanog, or Nestin. They are also identified in multiple solid tumors by means of the CSC’s specific cell surface markers and side population phenotype [85].

In CS, CSCs are commonly characterized by molecular markers such as aldehyde dehydrogenase (ALDH) and prominin-1 (CD133) [86]. ALDH is an enzyme that oxidizes aldehydes to carboxylic acids and allows cells to resist oxidative stress. Every cell expresses ALDH, however, cells with high ALDH activity have demonstrated enhanced tumorigenicity in several cancer cell types, characteristic of CSCs [87]. CD133 is a transmembrane glycoprotein, and its exact function remains to be elucidated, though it seems to be involved in membrane organization, cell differentiation, proliferation, and signal transduction. In addition, it might also have a role in apoptosis inhibition, and the upregulation of FLICE-like inhibitory protein (FLIP), leading to chemoresistance [88]. This cell-surface protein is known as an important driver of tumor progression and as a CSC marker [89].

Several studies identify CS CSC as ALDH+ and CD133+ cells and it was considered that the combination of ALDH+ CD133+ was the best marker to identify the tumor population enriched with the CSC phenotype [87,90]. Tirino et al. provided evidence of the presence of CSC in human primary bone sarcoma and demonstrated the possibility to use the CD133 marker for their identification [85]. They showed that CD133 was expressed in 21 fresh biopsies from bone and soft tissue sarcomas. After sorting cells, the CD133+ cells were able to reconstitute the original cell population, demonstrating the capacity of CSCs to dedifferentiate and rebuild tumor heterogeneity. Furthermore, they showed that the CD133+ cells were able to form tumorspheres. These spheres were positive for CD133 and the transcription factors OCT3/4, Nanog, Sox-2, and Nestin, which are involved in self-renewal and in the preservation of pluri-multipotency of normal stem cells. They demonstrated the ability of this cell population to differentiate into adipocytes and osteoblasts, supporting the fact that they originate from the mesenchymal stem cells of bone sarcomas. In addition, they showed in vivo that the cell population was capable of generating tumors in mice. These different evidences proved that the CD133+ sorted cells were CSCs, and, thus, CD133 is a useful marker for the identification of the CSC population. Greco et al. showed a significant correlation between ALDH activity and metastatic potential in ten patients with bone sarcoma, including CS. Moreover, they proved that bone sarcoma cells were sensitive to ALDH inhibition with disulfiram, involving a potential use of ALDH inhibition as a therapeutic strategy for radio and chemoresistant CS, although more investigations are required [87].

One of the main features of the CSC subpopulation is the overexpression of transcription factors such as SOX2 or OCT4, involved in the maintenance of stem cell phenotype in normal stem cells. In sarcomas, including CS, SOX2 has been found overexpressed in CSCs [91]. The same observations were made in relation to OCT4 in osteosarcomas and Ewing sarcoma, close models of CS. Menendez et al. introduced a system to monitor the transcriptional activity of SOX2 and OCT4 (SORE6) in CS patient-derive cell lines, in vitro and immunodeficient mice [91]. The system allows for isolating SOX2/OCT4 positive cells and thus analyzing the tumor-promoting CSC in sarcoma. They detected 20% of the SORE6+ cells, and this percentage was found increased to 40% in immunodeficient mice, which could be due to the elevation of the CSCs during tumor progression and adaptation to new microenvironments. They also showed that CSC-related genes, including SOX2, were overexpressed in the tumorsphere and enhanced during sarcoma progression. These results proved that SOX2 can be used as a CSC marker in sarcomas. Moreover, the system SORE6 is a good tool to evaluate the activity of antitumor drugs on CSC [91].

Another way to identify the CSC population is the so-called side population. CSC can evade treatment potentially through the increase in ATP-binding cassette (ABC) multidrug efflux transporters such as MDR1/ABCB1, BRCP1/ABCG2, and ABCB5 expression. The ability of cells to exclude DNA-binding dyes is measured. A side population appears as the cells expressing high ABC transporters exclude the dyes [84]. This was studied in osteosarcoma, however, it would be interesting to test this assay in the CS model.

## 3. Hadrontherapy and Combined Therapy of Chondrosarcoma

### 3.1. Hadrontherapy

Hadrontherapy presents several advantages over conventional therapy using X-rays: (1) it can accurately deliver a highly controlled dose of radiation to the tumor while sparing surrounding healthy tissue; (2) it is more effective at treating highly-resistant tumors; (3) the reduced exposure of the normal tissue makes it possible to reduce the length of the treatment and/or to increase the dose to the tumor [92,93].

#### 3.1.1. Physical Advantage of Hadrontherapy: The Bragg Peak

The first advantage of hadron is related to the physical characteristics of accelerated ions. Indeed, in the case of hadrons, as long as accelerated particles have a high speed (energy greater than 50 MeV/u), their ionizing effect on the tissues is relatively weak. Most of the energy takes place towards the end of their path. At that time, this deposit increases sharply over a distance of a few millimeters, then decreases rapidly. The profile describing the dose deposited as a function of the depth of the tissue crossed is called the Bragg peak [94,95]. The energy of the particle at the exit of the accelerator regulates the depth of penetration and the position of the maximum effect. Such energy deposit at the end of the path is greater (compared to the rest of the path where it is low) than with photons (for which the energy deposit is relatively linear). Consequently, a large part of the energy of the particle is deposited over a short distance. This ballistic quality makes it possible to reach more precisely the targets located in depth, and therefore to treat cancerous tumors that are inoperable or resistant to X or gamma rays, while better sparing the surrounding healthy tissues and/or the organs at risk. This property makes this type of radiation more precise than the photons (i.e., X-rays) used in conventional radiotherapy. Since the Bragg peak is too narrow to cover a tumor in the depth, during hadrontherapy, the Bragg peak can be spread (SOBP, spread out Bragg peak) with great dose homogeneity to cover the target volume. The technique consists of superimposing several single Bragg peaks from different beam energies [96].

It appears relevant to propose hadrontherapy for the treatment of chondrosarcoma in order to limit the irradiation of the tumor surrounding tissues such as brain tissue and the spinal cord when chondrosarcomas are situated at the base of the skull or pelvis and femur, and more generally, the cartilage which is a tissue that cannot renew. One of the phenomena that can cause damage to the surrounding healthy tissue is the appearance of the bystander effect. This phenomenon occurs through the secretion of stress factors by the irradiated cells that will impact nearby nonirradiated healthy tissues. The use of carbon ions has been shown to reduce the occurrence of this effect [57].

#### 3.1.2. Biological Advantage of Hadrontherapy: High LET and RBE

The second advantage of hadron is linked with the high ionization density at the end of the path of hadrons. This causes elevated DNA damage in cancer cells, which consequently will exhibit more difficulty in repairing themselves than the healthy cells located upstream. It is considered that this type of radiation is between 1.5 and 3 times more effective than a beam of photons (about 1.1 for protons) [93,97,98,99,100]. It corresponds to the “relative biological effects” (RBE). The ionization density of particles is directly related to the corresponding linear energy transfer (LET). High LET radiation (such as carbon-ion radiations) induces a greater proportion of DNA double-strand breaks (DSB) than low LET radiation (such as X-rays, and gamma-rays). These close DSBs participate in the formation of sites with multiple damages that are more complex, more numerous, and less easily repairable than X-ray induced damages (Figure 2).

In the particular case of chondrosarcoma, in vitro studies showed a higher efficiency of carbon ions on multiple chondrosarcoma cell lines [10,82,101,102]. The cellular effect was directly connected with the LET of carbon ions, inducing a prolonged block of irradiated cells in the G2 phase of the cell cycle and longer-lasting unrepaired cell damage [101,103]. Carbon ion radiations generated a positive regulation of several DNA repair genes (ATM, NBN, ATXR, XPC, XRCC1/2/3, ERCC1, XPC, and PCNA), activating a large range of DNA repair mechanisms [103].

Clearly, chondrosarcomas are particular tumors associated with hypoxia and cancer stem cells, conferring on them treatment resistance. So far, there is no study to prove the effectiveness of the use of ions on CS in a hypoxic environment.

However, in other models, the use of high LET ion radiations have already been shown to be more effective than X-rays in killing cancer cells under hypoxia [104,105]. It was already reported that carbon ions were more effective than X-rays in eliminating the subpopulation of chondrosarcoma cancer stem cells [82].

#### 3.1.3. State of Art of Hadrontherapy in Clinical Practice

There are two main ways to counteract CS radioresistance: (one) increasing the physical dose with organ-sparing irradiation methods, and proton beam irradiation is thus the technique of choice, or (two) increasing the biological dose using carbon irradiation in the absence or presence of radiosensitizing agents (i.e., chemotherapeutic).

The first way is becoming a standard. In fact, nowadays, high-dose proton therapy is considered the gold standard for the treatment of chondrosarcoma [106]. Many retrospective studies have been published with a very-high local-control rate for skull base chondrosarcoma and a low toxicity rate [107,108,109,110]. In one of this largest studies, long-term eight-year local-control rates for skull base chondrosarcoma, after high-dose proton therapy (70 Gy, 35 fractions), were 89.7%, with only 8% of patients developing a grade III-IV toxicity (hearing loss, radionecrosis, optic neuropathy, etc.) [107]. Among all studies, local control is associated with residue volume, age, and brainstem, or optic compression. Clinical reports about proton therapy for extra-cranial chondrosarcoma showed worse results with a local-control rate of about only 50% to 60% at five years [60,111,112]. If results for skull base chondrosarcoma are excellent for the treatment of extracranial locations, proton therapy needs to be challenged, and carbon irradiation, due to its high RBE, should be explored. Considering that chondrosarcomas are rare tumors and that worldwide accessibility to proton or carbon beam facilities remains difficult, building controlled randomized studies to compare conventional radiotherapy or proton therapy to carbon therapy is a hard challenge. Indeed, there are only two randomized trials currently recruiting patients. The first one, the French ETOILE trial (NCT02838602) [113], aims to compare the best radiotherapy, at the investigator site (IMRT or ideally proton therapy), with carbon therapy, delivered at CNAO. Poor prognosis, inoperable, or macroscopically incompletely resected (R2) radioresistant cancers, including chondrosarcoma ≥ grade II, are eligible, though skull base chondrosarcomas are excluded. On the opposite, the second one, a German trial, (NCT01182753) [114] is exclusively dedicated to good prognosis grade I-II chondrosarcoma of the skull base. Consequently, their main objectives are rather different. ETOILE Trial attempts to show an absolute improvement of the five-year PFS rate of 20% in favor of the experimental arm, while the German trial evaluates if the innovative carbon ion therapy in chondrosarcomas is not relevantly inferior to the standard proton treatment with respect to the five-year local progression-free survival.

Initial results of these studies are not expected for 5 or 10 years. However, there are a few retrospective studies or small prospective cohorts evaluating carbon irradiation, already published, with results close to the protons. Most of these studies report data from heterogenous series that include chondrosarcoma together with chordoma, or bone sarcoma. Data about chondrosarcoma patients have been extracted and are presented in Table 1. Only series with more than 10 patients are reported.

For skull base chondrosarcoma, the outcomes are similar to proton therapy with a high control rate (>90%) and a low grade III toxicity rate (≈10%) [115,116,117,119]. Except for headache and dizziness, all symptoms presented at baseline significantly decreased (−30% to −70%) after carbon beam irradiation, including fatigue, double vision, visual defect, and cranial nerve palsies [110]. For spinal and sacral chondrosarcoma, results are restricted to two studies and the results are disappointing with a 53% local-control rate at five years [118,120]. The toxicity rate remains safe with 10% rate of grade III late secondary effects, without an excess of myelopathy. Pelvis chondrosarcoma is more risky with 25% of late grade III toxicity, and up to 50% for hypofractionated schedule 70.4 Gy RBE/16fr [121].

Combining radiosensitizers with proton or carbon beams has not yet been explored in phase I studies. PARP inhibitor or IDH-inhibitor could be the first candidates [102,122].

### 3.2. Combined Approaches in Chondrosarcoma Control Strategy

#### 3.2.1. IDH Inhibitors

Targeting IDH mutations, one of the frequent mutations in chondrosarcomas could be a possible therapeutic solution. However, the first studies granted in chondrosarcoma, showed variable results when using an agent reversible the IDH mutation: AGI-5198 [123,124]. More recently, studies carried out with CRISPR/Cas9 technology [125] and with a new IDH mutation inhibitor, the DS-1001b [126], demonstrate the importance of this mutation in the tumorigenicity of chondrosarcoma, and therefore the interest in targeting this mutation for chondrosarcoma treatment.

#### 3.2.2. PARP Inhibitors

PARPs are proteins involved in DNA repair systems. The major protein, PARP1, is mainly involved in base excision repair (BER), a single-stranded repair system [127]. The interest in the use of PARP inhibitors is found mainly in the case of BRCA1 and BRCA2 mutated tumors [128]. Indeed, these BRCA proteins are involved in double-stranded DNA repair systems by homologous recombination. The use of PARP inhibitors makes it possible to maintain single-strand breaks in tumor cells, which will lead to double-strand breaks during replication. Since homologous recombination is not effective in the case of cells mutated for BRCA, the very toxic double-strand breaks will persist at the cellular level and lead to cell death [128].

PARP inhibitors are mainly efficient in dividing cells which allows a low effect on organs at risk, especially in combination with localized irradiation [129]. Indeed, these molecules are interesting in combination with radiotherapy since they would make it possible to maintain the DNA breaks induced by the different qualities of irradiation (Figure 1). This effect has been demonstrated in the case of chondrosarcoma, with radiosensitization of chondrosarcoma cells against the effect of X-rays, protons, and carbon ions, with a very marked effect in association with protons [102]. Recently, studies have demonstrated that IDH mutations can lead to a suppression of homologous recombination, thus inducing a BRCAness phenotype [130,131]. Under these conditions, PARP inhibitors can be considered as a therapeutic solution since they can lead to synthetic lethality. However, in the case of chondrosarcoma, IDHm and IDHwt cells showed variable sensitivities to the PARP inhibitor talazoparib, independently of their mutation. Also, the reversion of the IDH1 mutation by AGI-5198 does not modify the response to talazoparib [132]. Thus, treatment with PARP inhibitors alone, in the case of CHS, does not seem to be relevant. However, the use of these inhibitors sensitizes chondorsarcoma cells to radiotherapy treatments [102], as well as to temozolomide [132]. Interestingly, these sensitizations seem more effective on IDHm cells [132].

#### 3.2.3. Targeting the Hypoxic Microenvironment

As mentioned above, hypoxia plays an important role in the radioresistance of chondrosarcoma tumor cells. Indeed, the low oxygen tension directly induces the reduction of the effects of X-rays [63]. Also, the stabilization of the HIF-1α and HIF-2α factors and the activation of their signaling pathways allows tumor progression via angiogenesis [66,68], the induction of autophagy [64], the expression of antiapoptotic factors [70], or the induction and stabilization of CSCs [71]. This is why the inhibition of these main factors involved in hypoxia signaling could be a good alternative to radiosensitize tumor cells (Figure 1).

It was recently demonstrated that inhibition of HIF-2α could induce a reduction in the invasive properties of chondrosarcoma cells [26]. In addition, when using a specific HIF-2α antagonist, TC-S7009, the authors were able to demonstrate a chemosensitization of chondrosarcoma cells to cisplatin and doxurubicin. This effect has not been explored in CS CSCs. HIF-2α inhibitors are being studied in the clinical setting, namely PT2385 [133] and PT2977 (NCT02974738). Nevertheless, these molecules have not yet been tested in the case of CS, nor in association with irradiation, however, they have the potential to be a good option in the management of chondrosarcoma.

#### 3.2.4. Targeting CSCs

Therapeutic strategies are needed in order to target both the CSCs and the nonstem cancer cells to avoid therapy-induced CSCs, leading to metastasis and tumor relapse. Numerous reports proposed potential targets against CSCs in solid tumors such as targeting cell surface markers, signaling pathways, CSC niches, or inhibitors to overcome drug resistance [83].

The evidence indicated that the mTOR pathway might have an important role in CSCs maintenance. A combination of RX and carbon ions with drug treatments was used on the CS cell line CH2879. They showed that rapamycin, an mTOR pathway inhibitor, combined with miR-34, a tumor-suppressive micro-RNA, associated with the regulation of stem-like cells in solid tumors, could overcome CSC-associated radioresistance. As such, a higher decrease in tumorsphere formation and (aldehyde dehydrogenase) ALDH+ CSCs was observed in cells treated with rapamycin and miR-34 combination, compared to individual treatments. In addition, it was also demonstrated that the combined treatment improves carbon-ion therapy at a lower dose than used in the case of X-ray, suggesting that the risks of relapse and metastasis might be decreased and the environing tissues could be better preserved [82]. Other studies reported proline-rich polypeptide 1 (PRP1) as a potential therapeutic agent in CS to target CSCs [116,120,134]. PRP1 is known to have cytotastic, antiproliferative, immunomodulatory, and tumor suppressor properties, and is an mTOR inhibitor (Figure 1). It was shown that PRP1 treatment on the JJ012 cells’ monolayer and the 3D spheroid model significantly decreased the ALDH high CSC population [134,135]. A combination of disulfiram and Cu^2+^ (DSF/Cu) has also been proven to target CSCs. Wang et al. studied the complex DSF–CU as a radiosensitizer on the SW1353 cell line and the ability of the complex to target CSC by analyzing the ALDH+/CD133+ level, tumorspheres, stem marker expression, and the inhibition of the stem feature in mice. They highlighted the increase of CSCs after irradiation by tumorsphere formation and the increased level of stem transcription factors HER2 and SLUG. DSF–Cu association allowed for the elimination of almost all of the tumorspheres in vitro, and the decrease of ALDH+/CD133+ cells. In vivo, DSF–Cu decreased the tumor volume and enhanced survival in mice. The antitumor activity of this complex could enable the increase of the therapeutic index of radiotherapy [90].

Although more and more strategies are emerging in order to target CSCs, there are still only a few studies in CS. Therefore, more investigations using CS are required to better identify the different mechanisms and the responses to therapies.

## 4. Conclusions

Chondrosarcoma is resistant to conventional antineoplastic treatments, owing to its particular microenvironment. The hypoxic environment directly induces chemo- and radioresistance. It may also promote cell survival and differentiation into CSCs, which are naturally more resistant to chemo- or radiotherapy.

The development of hadrontherapy seems a relevant treatment alternative. Preliminary results indicate a better antitumoral effect. In addition, the use of specific molecules such as PARP, HIF, or IDH inhibitors, in combination with hadrontherapy, remains a promising strategy to improve local control and overall survival.

## Figures and Tables

**Figure 1 cancers-15-01962-f001:**
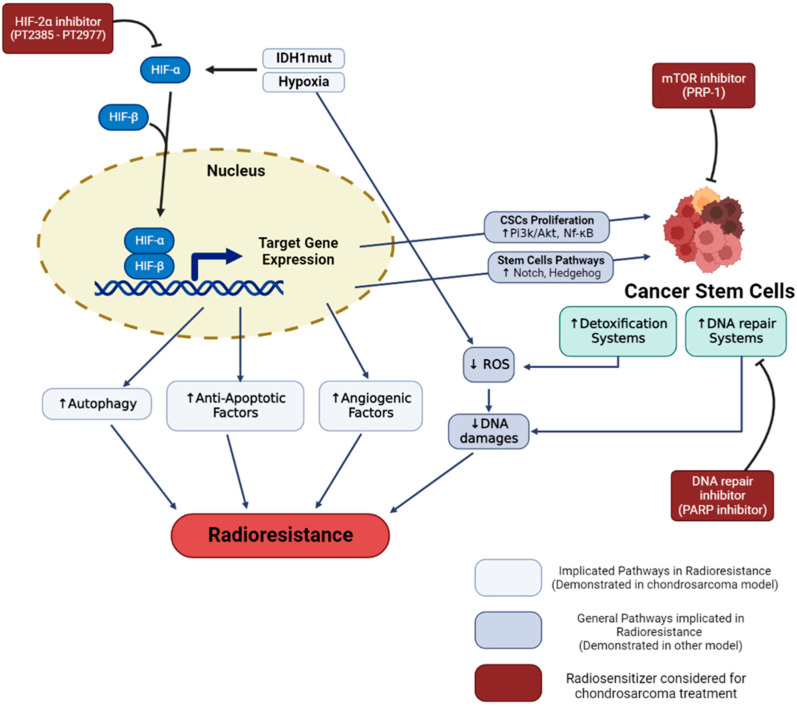
Hypoxia and cancer stem cells radioresistance in chondrosarcoma. Hypoxia and IDH-1 mutation lead to HIF-1α and HIF-2α stabilization. These regulated pathways enhance cell radioresistance and increase the proportion of cancer stem cells. Specific inhibitors (in red), such as HIF-2, PARP, and mTOR inhibitors demonstrated a capacity to reverse cell resistance and are considered in order to radiosensitize chondrosarcomas.

**Figure 2 cancers-15-01962-f002:**
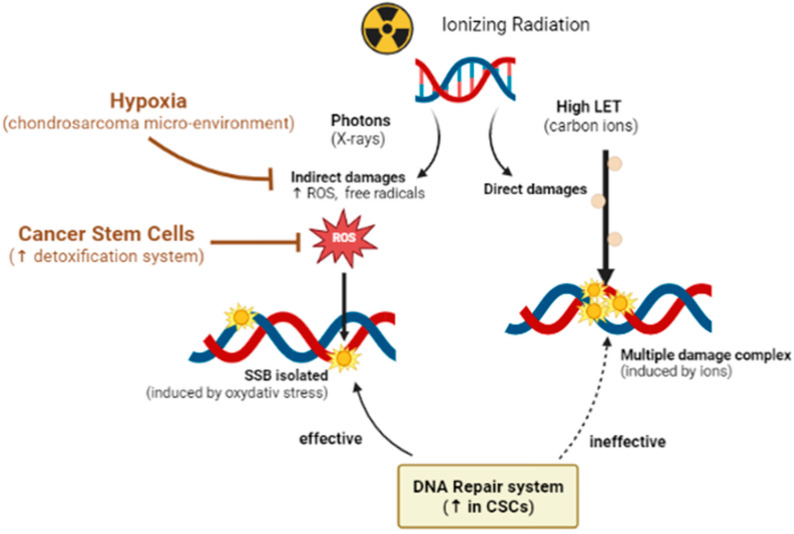
Benefits of High LET irradiations. In the case of ionizing radiations, X-rays induce mainly indirect damages, through the formation of free radicals and reactive oxygen species (ROS), which generate isolated DNA damages. As observed in the case of chondrosarcoma, hypoxia can reduce such damages. In case of high ROS levels, an enhancement of detoxification systems is triggered in cancer stem cells. High LET radiations, such as carbon ions, induce more direct complex damages, named localized multiple damage, independently of the oxygen level and more difficult to repair for cells.

**Table 1 cancers-15-01962-t001:** Clinical studies involving chondrosarcoma patients (>10 patients).

Reference	Patients (*n*)	Indication	Dose	Efficacy	Tolerance
[115]	23 with low-grade chondrosarcoma	R2 or Biopsied only patients	60 GyRBE using a weekly fractionation of 7 × 3.0 GyRBE.	100% local-control rates at 3 years	9% of grade III late-effects
[116]	54 with grade I–II skull base chondrosarcoma	R2 or Biopsied only patients (including recurrent tumors)	60 GyRBE using a weekly fractionation of 7 × 3.0 GyRBE.	local-control rates were 96.2% at 3 years and 89.8% at 4 years	2% of grade III late effects
[117]	79 patients (64% grade I, 35% grade II, 1% grade III) with skull base chondrosarocma	Recurrences, R2, or Biopsied only patients	60 GyRBE at 3 GyE per fraction	CI, 88.8–100%) and 89.8% at 4 years	No grade III effects reported
[118]	73 patients (20% grade I, 70% grade II, 5% grade III, 5% dedifferenciated). Extracranial only.	Biopsied only patients (75%), Recurrence or metastatic (25%)	70.4 GyRBE, 16 fractions, 4 consecutive days a week, 4 weeks	5-year local-control, overall survival, and disease-free survival rates were 53%, 53%, and 34%	11% of grade III late effects
[119]	16 patients with skull base chondrosarcoma (75% of grade II–III)	R2 or Biopsied only patients	70.4 GyRBE, 16 fractions, 4 consecutive days a week, 4 weeks	3-year LC rate of 94%	12.5% of grade III late effects
[120]	21 patients with chondrosarcoma. Extracranial only	Not available	73.6 Gy(RBE) delivered in 16 fractions (4 fractions per week)	not available	<5% of grade III late effects

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
