# Peer review of "Chondrosarcoma Resistance to Radiation Therapy: Origins and Potential Therapeutic Solutions"

_cancers, 2023, doi:10.3390/cancers15071962_

Round 1

Reviewer 1 Report (Previous Reviewer 3)

In this review, the Authors aimed to describe the different forms of chondrosarcoma resistance and how hadrontherapy, combined with other treatments involving targeted inhibitors, could help to better treat this type of cancer.

The topic is interesting. However, the paper is disorganized. 

Despite ameliorated, much imprecise information are reported. 

The Authors try to reach conclusions based on results obtained by previous studies. This is not an objective of such a narrative review.

I still believe that a narrative review on current treatments of Chondrosarcoma (including radiation therapy) would be more useful. The treatment of ChS should be differentiated among tumor grade. Grade 1 ChS cannot be treated as a dedifferentiated one.

At this point, I would also limit introduction on the treatment of high grade ChS, limiting information to the essential ones for the objective of this review.

Methodologically, it is inappropriate to draw conclusions based on previous reports. A narrative review should only report previously published data. 

Author Response

My co-author and I have carefully considered the helpful suggestions made, and have revised the manuscript accordingly.

We thank the reviewer 1 whose comments helped us to substantially improve the manuscript.

As suggested, we added a new chapter summarizing the current treatments of chondrosarcoma, as a function of the grade and of the location. In addition, we specified that the review is now specifically focused on the treatment of high grade chondrosarcoma.

In addition, the structure of the paragraph was modified and the text revised, especially in relation with the current treatments.

We tried to modify some of our interpretations in order to be less direct on the conclusions reached.

Reviewer 2 Report (Previous Reviewer 2)

Thanks for your great wort. I think this form could be published

Author Response

Thank you for your comments

Reviewer 3 Report (Previous Reviewer 1)

no comments

Author Response

Thank you for your comments

Round 2

Reviewer 1 Report (Previous Reviewer 3)

I appreciate the efforts the Authors made in the attempt to ameliorate their paper.

However, information on the indications to radiotherapy in high grade ChS are confusing and might be misleading. Would the Authors irradiate a grade 2 ChS marginally resected?

Author Response

Thank you for your comment.

In agreement with the first request of reviewer 1, we modified the text related to chondrosarcoma treatment. We added a new paragraph (1.3), in which we detailled the treatment in regards with chondrosarcoma grade and tumor surgery. The explanation was writen in agreement with ESMO guideline.

Regarding the last comment of reviewer 1, indeed, a grade II chondrosarcoma marginally resected will be irradiated, after surgery.

We hope that this part of the manuscript is now acceptable.

This manuscript is a resubmission of an earlier submission. The following is a list of the peer review reports and author responses from that submission.

Round 1

Reviewer 1 Report

This is a comprehensive review of the critical topic. Well written and gives excellent insight into the problem with chondrosarcoma. 

I do have a few suggestions:

1. Genomics has been very much ignored in the manuscript, only SOX2 and OCT4 monitoring is mentioned. However, several studies (PMID: 29250102, 29050494) suggest that, at least in the case of osteosarcoma, the transcriptomic profiles could show the individual variability of the sarcoma and therefore provide some extra insight into the therapeutic response.

2. Biological therapy is not discussed at all. As some opportunities are emerging (PMID: 29361725) then this could be discussed even briefly.

Reviewer 2 Report

Dear authors,

Thanks for your great work. There are some comments that I hope you can address.

1.IDH1/2 mutations were found in 71% of conventional chondrosarcomas and 57% of dedifferentiated chondrosarcomas. IDH mutation is among one of the promising therapeutic targets in chondrosarcoma.

Please clarify the importance of IDH mutation in two aspect of chondrosarcoma. First, How IDH1 changed the microenvironment of chondrosarcoma. Second, please comment

the relationship of IDH mutation and radioresistance in chondrosarcoma. 

2.In the era of precision medicine era, please comments whether there are some molecular subtype resistant to radiotherpy.

3.IDH1/2 mutations induce an HR defect that renders tumor cells sensitive to PARP. PARP inhibitor combine with radiotherpy is promising in IDH mutation tumors.

4.In page 4 “Hif-2α factors” should be “HIF-2α”.

Reviewer 3 Report

In this review, the Authors aimed to describe the different forms of chondrosarcoma resistance and how hadrontherapy, combined with other treatments involving targeted inhibitors, could help to better treat this type of cancer.

The topic is interesting, but the paper is disorganized and it doesn't sound scientific at all.

The Authors try to reach a conclusion based on results obtained by previous studies. This is not an objective of such a narrative review.

Moreover, much imprecise and inappropriate information are reported (eg. "Current chemotherapeutic treatments are based on the results previously obtained from the treatment of osteoarthritis"....what's the meaning of this sentence?)

Probably, a narrative review on current treatments of Chondrosarcoma, maybe differentiating them among tumor grade, would be more useful.

grade 1 Che cannot be treated as a dedifferentiated one.